# Co-Doping Effect on the Optical Properties of Eu^(2+/3+)^ Doped in BaSiO_3_

**DOI:** 10.3390/ma15196559

**Published:** 2022-09-21

**Authors:** Purevdulam Namkhai, Kiwan Jang

**Affiliations:** Department of Physics, Changwon National University, Changwon 51140, Korea

**Keywords:** BaSiO_3_, photoluminescence, emission efficiency, pc-white LED phosphor, divalent europium

## Abstract

To investigate the effect of co-doping on the optical properties of Eu^(2+/3+)^ doped in Ba_0_._98_SiO_3_:0.02Eu, the series of Ba_0_._96_SiO_3_:0.02Eu, 0.02R^+/3+^ (R^+^ = Li^+^, K^+^ or Na^+^, R^3+^ = La^3+^ or Y^3+^) phosphors were synthesized using a solid-state reaction method. The excitation efficiency due to the charge transfer band (CTB) was enhanced via co-doping of R^+^ and the emission intensity due to Eu^3+^ was thus increased by 3.7 times compared with that of the single-doped Ba_0_._98_SiO_3_:0.02Eu^3+^. However, the co-doping of R^+^ does not increase the emission intensity of Eu^3+^ via the direct ^7^F_0_→^5^L_6_ excitation of Eu^3+^, but rather decreases it. On the other hand, the emission intensities due to Eu^2+^ were decreased via the co-doping of R^+^ but increased via the co-doping of La^3+^. The present work reveals that the optical properties of Eu^3+^ or Eu^2+^ doped in BaSiO_3_ depend not on the charge state (+ or 3+) of the co-doped ions, but on the co-doped element itself.

## 1. Introduction

The photoluminescent properties of rare-earth (RE) ions have attracted considerable interest because of their potential applications as phosphors, white light-emitting diodes (WLEDs), and lasers, etc. When doped in inorganic host materials, RE ions energetically prefer the trivalent state in hosts. Some RE ions, such as Eu, Sm, and Yb, where Yb^2+^ is more stable than Yb^3+^, are doped with divalent atoms in some hosts. Specifically, Eu^2+^-doped phosphors exhibit a broad emission band that varies from ultra-violet to red, depending on the host lattice, and some of Eu^3+^-doped phosphors are used as a typical red phosphor. Thus, europium-based luminescence materials are extensively used in solid-state lighting, solar cells, red phosphors, and WLEDs.

Silicates are widely used as phosphor hosts due to their merits, such as their high thermal stability, the high energy efficiency of the silicate phosphors, and the inexpensive nature of the raw materials. Most often, the europium ions in a host are substituted into alkaline earth metals such as Ba^2+^, Sr^2+^, or Ca^2+^. When Eu^3+^ ions are doped in silicate hosts such as BaSiO_3_, Ca_2_SiO_4_, and Sr_2_SiO_4_ through heat-treatment in air, there should be a charge balance mechanism involved in replacing A^2+^(A^2+^ = Ba^2+^, Ca^2+^, and Sr^2+^) with Eu^3+^. In the case of BaSiO_3_, in which Eu^3+^ replaces the Ba^2+^ site based on the ionic radius, negatively charged barium ion vacancies (V_ba_) are created for charge compensation. It is known that the creation of such defects leads to a deterioration in the optical performance of the synthesized phosphors. Jiang et al. have investigated the influence of Li^+^ co-doping on the photoluminescence of Eu^3+^ doped in an SrWO_4_ host [1]. Li et al. also studied the effect of Li^+^, Na^+^, and K^+^ on the PL enhancement of Tb^3+^ doped in CaMoO_4_ [2]. In many cases, it has been reported that the co-doping of alkali-metal ions or Y^3+^ or La^3+^ enhances the emission intensity of RE ions doped in host materials [3,4,5,6,7]. However, it has also been reported that the emission intensity decreased in ZnTa_2_O_6_:Pr^3+^ co-doped with alkali-metal ions [8]. The Eu^3+^ ions doped in a BaSiO_3_ host will be located along with the Ba^2+^ site because the ionic radius of Eu^3+^ is similar to that of Ba^2+^. Meanwhile, the BaSiO_3_:Eu^2+^ phosphors are mostly prepared via the heat-treatment of BaSiO_3_:Eu^3+^ in a reducing atmosphere at high temperature. In this reduction process, the V_ba_ defect created for charge compensation in BaSiO_3_:Eu^3+^ will lose its charge compensation function. To date, to the best of the authors’ knowledge, there have been no reports on the effect of V_ba_ on the optical properties of Eu^2+^ doped in BaSiO_3_.

In this article, we report on the co-doping effects of alkali metal ions and La^3+^ (or Y^3+^) on the optical properties of BaSiO_3_:Eu^(2+/3+)^ fabricated via a solid-state reaction (SSR) method. The effects of Li^+^, K^+^, Na^+^, La^3+^, and Y^3+^ co-doped with Eu^(2+/3+)^ on the crystalline phase and the optical properties of BaSiO_3_:Eu^(2+/3+)^ are discussed. The results showed that the doping of Li^+^, K^+^, Na^+^, La^3+^, and Y^3+^ can significantly affect the optical properties of Eu^(2+/3+)^ but the element used for the co-doping ion has a greater effect on the optical properties of Eu^(2+/3+)^ than the charge state (+ or 3+) of the co-doping ion.

## 2. Materials and Methods

A series of Ba_0_._98_SiO_3_:0.02Eu, Ba_0_._96_SiO_3_:0.02Eu, and 0.02R (R = Li, K, Na, La, and Y) phosphors were prepared using a conventional solid-state reaction. The BaCO_3_ (99+%), SiO_2_ (particle size 5~20 nm, 99.5%), Eu_2_O_3_ (99.99%), K_2_CO_3_ (99+%), Li_2_CO_3_ (99+%), Li_2_O (97%), Na_2_CO_3_ (99+%), LaCl_3_ (99.99%), and Y_2_O_3_ (99.99%), purchased from Merck, were used as a raw materials but SiO_2_ was used 20% more to obtain a single-phase BaSiO_3_ sample. The raw materials were thoroughly mixed in an agate mortar and subsequently fired in air at 1100 °C for 4 h to synthesize the Eu^3+^-doped BaSiO_3_. The BaSiO_3_:Eu^2+^ samples were prepared by sintering the BaSiO_3_:Eu^3+^ samples again under a reduction atmosphere (95% N_2_ + 5% H_2_) at 1200 °C for 6 h. The specific formulations of the studied samples are shown in Table 1.

The crystal phase of the obtained samples was identified using an X-ray diffractometer (X’Pert PRO MPD-3040, Malvern Panalytical, Malvern, UK, CuK_α1_, λ = 1.5406 Å) operating at 40 kV and 30 mA with a scan speed of 0.02°/sec. The photoluminescence excitation (PLE) and emission (PL) spectra were measured using a fluorescent spectrofluorometer (JASCO FP-8500, JASCO, Tokyo, Japan) equipped with an integrating sphere (ISF-834, JASCO, Tokyo, Japan). All measurements were performed at room temperature.

## 3. Results and Discussion

Figure 1 shows the X-ray powder diffraction (XRD) patterns of the studied samples. The XRD patterns of all samples were relatively well matched with pure BaSiO_3_ (JCPDS-702112) within the accuracy of our XRD equipment, indicating that the studied samples were single-phased BaSiO_3_, which crystallizes in an orthorhombic structure and has a space group of P2_1_2_1_2_1_ with lattice constants of a = 0.4580 nm, b = 0.5611 nm, and c = 1.2431 nm, α = β = γ = 90° [7]. This also means that the activator Eu^(2+/3+)^ ions and co-doping ions R^+/3+^ which were small-doped in the samples, did not cause any significant changes in the crystal phase of BaSiO_3_. Miscellaneous minor peaks were marked with a plus (+) and an asterisk (*). The XRD patterns of samples 8–14, prepared under a reducing atmosphere, were identical to those of samples 1–7 because there was no change in crystallinity even if the secondary heat treatment was performed under a reducing atmosphere. The XRD patterns of the samples prepared under a reducing atmosphere are thus not presented in Figure 1.

Based on the data provided by the Cambridge Crystallographic Data Center, Si^4+^ ions are located at the center of a tetrahedron coordinated with four O_2_^−^ ions. The [SiO_4_]^2−^ tetrahedrons form a chain running parallel to the *c*-axis by sharing corners and the Ba^2+^ ions are accommodated in the tunnels between chains [9,10].

In our previous work [11], the crystal structure of BaSiO_3_ was found to be highly dependent on the particle size of the SiO_2_ used as a raw material. Even though this result was not presented in this paper, it was not possible to fabricate samples not having the minor second phase with SiO_2_, the particle size of which was greater than 500 nm. This means that the characteristics of raw materials having the same molecular formula can highly affect the crystal structures of the samples prepared under the same preparation conditions. Specially, the optical properties of BaSiO_3_:Eu^2+^ are highly sensitive to the presence or absence of the minor second phase in the crystal structure, even though this is not true of other materials. The main reason that previously reported results are slightly different may be due to the difficulties involved in achieving single-phase BaSiO_3_:Eu^2+^ fabrication without having some minor peaks [10,12,13].

The crystal structure of BaSiO_3_ has been detailed in other works [10,14], where a Ba site and six oxygen atoms bond together in the BaSiO_3_ unit cell. Since Ba^2+^ ions (0.135 nm) are closer to Eu^3+^ ions (0.0947 nm) than Si^4+^ ions (0.0400 nm) in terms of their ionic radius, Ba^2+^ is expected to be replaced by Eu^3+^ [7]. Assuming that Ba^2+^ is expected to be replaced by Eu^3+^, an imbalance of charges is generated in the sample, resulting in the creation of a negatively charged defect, V_Ba_. To solve this problem, charge compensators such as Li^+^, K^+^, or Na^+^ were co-doped with Eu^3+^ and the effect of charge compensation on the emission of Eu^3+^ has been reported [7]. Yang et al. reported that doping with Na^+^ and K^+^ ions has a drastic effect on the improvement of the excitation and emission of Eu^3+^ [7]. However, through this work, we found that doping with Y^3+^ ions also has the same effect as that obtained with Na^+^ and K^+^ ions. This result means that the improvement of the excitation and emission of Eu^3+^ could be determined by the species of co-doping ions, rather than the charge state of the co-doping ions, and also shows that this process may not be explained with charge compensation alone, as described so far.

Figure 2 illustrates the excitation and emission spectra of Ba_0_._96_SiO_3_:0.02Eu^3+^, 0.02R^+/3+^ samples. The PLE, measured by monitoring the strongest Eu^3+^ emission at 612 nm, is shown in Figure 2a. When Eu_2_O_3_, which is an optical active material, was mixed with other host materials and then heat-treated in air, the Eu ions were doped into Eu^3+^ in the host material due to its chemical stability. The PLE spectra exhibit a wide band known as a charge transfer band (CTB), where the electrons come from the entirely filled 2p orbitals of O^2-^ ions and are transferred to the partially filled 4f orbitals of Eu^3+^. The CTB ranged from 200 nm to 290 nm, with the highest peak centered around 228 nm. Lingxiang Yang et al. reported that the CTB peaked around 246 nm [14]. This 18 nm difference in the CTB peak position may be attributed to the crystallinity of the studied samples. The sample studied by Lingxiang Yang et al. exhibited relatively many miscellaneous peaks, which were ascribed to the BaSi_2_O_5_ phase (JCPDS#26-0176), compared with our samples.

Based on our previous study, we knew that the optical properties of Eu^(2+/3+)^ doped in BaSiO_3_ were more highly dependent upon the crystallinity of the host materials, compared to other host materials. Several narrow bands from 300 nm to 550 nm, due to the intra-4f transitions of Eu^3+^ ions, were also observed. Since the transitions of Eu^3+^ were very stable, and are almost certainly not affected by the lattice structure or neighboring anion ligands of Eu^3+^, the excitation peaks were almost same for all samples. However, in the case of the CTB, things were quite different. It has been reported in the literature that the CTB depends greatly on the lattice structure of the matrix in which Eu^3+^ was introduced [15]. For example, the CTB peak position for Sr_2_SiO_4_:Eu^3+^ is 293 nm and it was 250 nm for the Gd_2_O_3_ [16]. It is thus reasonable to assume that the 18 nm difference in the CTB peak position may be attributed to the crystallinity of the studied samples.

One interesting aspect of our results is that the excitation efficiency due to the CTB in sample 5, co-doped with Na^+^ ions, was enhanced by 3.7 times compared with that of the single-doped sample 1. The CTB excitation efficiency of sample 7, co-doped with Y^3+^, was two times higher than that of sample 6, co-doped with La^3+^. Furthermore, The CTB excitation efficiency of samples co-doped with Li^+^ or K^+^ was twice as high as that of the single-doped sample 1. These results mean that the co-doped R^+/3+^ ions act as a sensitizer and can thus contribute to the enhancement of the emission intensity due to Eu^3+^. It is thus relatively reasonable to conclude that the CTB excitation efficiency does not depend greatly on the charge state of the co-doping element, but rather on the species of co-doping ions.

Upon the excitation of CTB’s 230 nm region, as shown in Figure 2a, the ^5^D_0_→^7^F_2_ maximum transition intensity of Eu^3+^ for sample 5, co-doped with Na^+^, was enhanced by a factor of 3.7 times compared with sample 1. In the case of samples 2 and 3, which were co-doped with the same Li^+^, the ^5^D_0_→^7^F_2_ transition intensity of sample 2 was stronger than that of sample 3. We have thus shown through this study, for the first time, that the emission efficiency of Eu^3+^ is dependent upon the raw material of the co-doping ions, rather than the charge state of the co-doping ions. Figure 2c shows the emission spectra obtained at an excitation wavelength of 393 nm, which represents a ^7^F_0_→^5^L_6_ excitation of Eu^3+^. The emission intensities of samples 2, 3, and 6, co-doped with Li^+^ or La^3+^ ions, were stronger than that of sample 1 but that of sample 4, co-doped with K^+^, Na^+^ and K^+^, was much weaker than that of sample 1.

Figure 3 shows the strongest Eu^3+^ emission intensity that was obtained at 612 nm when excited by 230 nm and 393 nm, and it also shows the difference in the excitation efficiency through the CTB and the direct ^7^F_0_→^5^L_6_ excitation of Eu^3+^. When excited by CTB’s 230 nm region, the results clearly showed that the excitation efficiencies of sample 5, co-doped with Na^+^, and sample 4, co-doped with K^+^, were much better than those of other samples. However, the excitation efficiency of sample 4, excited by 393 nm, which represents ^7^F_0_→^5^L_6_ excitation of Eu^3+^, was the lowest among the studied samples.

Figure 4a–d shows the PLE and PL spectra of the studied BaSiO_3_:Eu^2+^ phosphors (samples 8–14). The PLE spectra, measured by monitoring the 577 nm emission intensity, are shown in Figure 4a. The PLE spectra cover a very broad region, from 200 to 450 nm, with two peaks of 280 nm and 340 nm. The broad excitation covered the UV/vis range widely, which matches well with commercial LED chip emissions, so this phosphor is suitable for solid-state lighting based on an n-UV LED chip. The broad excitation band is commonly ascribed to the crystal-field split 4f^6^5d^1^ configuration of Eu^2+^ ions. It is well known that the 4f^6^5d^1^ electron state is not shielded from anion ligands in the compounds. It has thus been suggested that the 4f^6^5d^1^ electron state has a strong interaction with neighboring anion ligands in the BaSiO_3_ host, which results in a broad excitation band, as shown in Figure 4a. Since samples 8–14 were actually the same as samples 1–7, which were heat-treated in air, except for the fact that samples 8–14 were heat-treated in a reducing atmosphere, and the ionic radius of Eu^3+^ (0.0947 nm) was smaller than that of Ba (0.135 nm), it was expected that Eu^2+^ ions in sample 8–14 would replace the Ba sites of BaSiO_3_. The excitation spectrum shown in Figure 4a has two peaks at 280 nm and 340 nm, which corresponds to the crystal-field splitting of the 5d levels in the excited 4f^6^5d^1^ configuration of the Eu^2+^. The excitation spectra for samples 8–14 show a similar pattern, even though the relative excitation intensity over the whole excitation band is slightly different.

Figure 4b shows the room-temperature PL spectra of samples 8–14 excited at 340 nm, in which the PL features of Eu^2+^ showed slight changes in peak position depending on the co-doping ions. Since only one Ba site exists crystallographically in BaSiO_3_, a single broad emission band from 450 nm to 750 nm with the peak location at 577 nm was expected, which was attributed to the typical 4f^6^5d^1^→4f^7^ transition. This single and symmetrical broad emission band indicated that the Eu^2+^ ions were replaced into the only Ba^2+^ site in BaSiO_3_, as expected. The full width at half maximum (FWHM) of the emission spectrum was about 131 nm, which is broader than the common values of FWHM (50–100 nm) of Eu^2+^ in most phosphors [3,12,17] and is also broader than that (119 nm) of the well-known yellow-emitting Y_3_Al_5_O_12_:Ce^3+^ phosphor [18]. This broad emission may indicate that the interaction of Eu^2+^ with the ligands of the BaSiO_3_ host was strong and thus it may also be suitable as a warm white light source using an n-UV Led chip.

We also found the emission due to Eu^3+^ ions when samples 8–14 were excited at 280 nm was efficient in exciting Eu^3+^ ions. This implies that not the all Eu^3+^ ions present in samples 1–7 were completely reduced into Eu^2+^ via heat treatment under the mixed nitrogen (90%) and hydrogen (10%) atmosphere. Co-doping ions such as Li^+^, K^+^, and Na^+^ were effective in enhancing the emission of Eu^3+^ via the excitation of the CTB (refer to Figure 2b) but they decreased the emission intensity of Eu^2+^ ions, as shown in Figure 4b. Based on the radii of the compound elements, all Eu^3+^ ions were expected to replace the Ba^2+^ sites of BaSiO_3_ due to the occurrence of the charge imbalance. To solve this charge imbalance effect, charge compensators such as Li^+^, K^+^, and Na^+^ ions were co-doped with Eu^3+^ ions. When Eu^3+^ and R^+^ (R^+^ = Li^+^, K^+^ and Na^+^) were co-doped, two kinds of defects were created, since one Eu^3+^ and one R^+^ occupied two Ba^2+^ sites in BaSiO_3_:Eu^3+^, R^+^ (samples 2–5). Considering that the ionic sizes of cations varied with their coordination numbers (CN), we expected that Eu^3+^ and R^+^ ions would prefer to replace the Ba^2+^ (CN = 8, r = 1.42Å) in BaSiO_3_ because the ionic radius of Eu^3+^ (CN = 6, r = 1.07~1.087 Å) was closer to that of Ba^2+^ and the Si^4+^ sites with an ionic radius of 0.42 Å were too small for Eu^3+^ to occupy [7,9,19]. The ionic radii of Li^+^, Na^+^, and K^+^ ions are also known to be 0.9, 1.16, and 1.52 Å, respectively [19,20].

Ba_0_._96_SiO_3_:0.02Eu^2+^, 0.02R^+/3+^ (samples 8–14) phosphors were prepared by carrying out a second heat-treatment of Ba_0_._96_SiO_3_:0.02Eu^3+^, 0.02R^+/3+^ samples (samples 1–7) in a mixed nitrogen (90%) and hydrogen (10%) atmosphere to reduce Eu^3+^ into Eu^2+^. When Ba_0_._96_SiO_3_:0.02Eu^3+^, 0.02R^+/3+^ phosphors were heated in the mixed nitrogen (90%) and hydrogen (10%) atmosphere, hydroxyl bonds were formed, which seemed to play an important role in the reduction of Eu^3+^ ions. We assume that these hydroxyl bonds may have reacted with the negatively charged barium ion vacancies (V_ba_) which were generated due to the charge imbalance when Eu^3+^ replaced the Ba^2+^ site in Ba_0_._96_SiO_3_:0.02Eu^3+^, 0.02R^+/3+^, resulting in the decreased PL intensity of Eu^2^. However, more experimental and theoretical studies are needed to reveal the mechanism underlying the decrease in PL intensity due to Eu^2+^.

Figure 4d shows the room-temperature PL spectra of samples 8–14 excited at 254 nm, which belonged to the CTB and was also efficient to excite Eu^3+^ ions. As shown in Figure 4c, the PL intensity of sample 13, co-doped with La^3+^, was stronger than that of samples 9–12, co-doped with R^+^. Furthermore, the PL intensity of sample 13 was stronger than that of sample 14, which was co-doped with Y^3+^. Figure 2 shows that the PL intensity of the ^5^D_0_→^7^F_2_ transition of Eu^3+^ for sample 6, co-doped with La^3+^, was weaker than that of the ^5^D_0_→^7^F_4–7_ transition but the PL intensity of the ^5^D_0_→^7^F_2_ transition of Eu^3+^ for samples 1–7 (except for sample 6) was stronger than that of the ^5^D_0_→^7^F_4–7_ transition. In general, it is known that the emission intensity due to the ^5^D_0_→^7^F_4–7_ transition is weaker than the emission intensity caused by the ^5^D_0_→^7^F_2_ transition because Ω_4_ is much smaller than Ω_2_ according to the Judd–Ofelt theory [20]. However, depending on the symmetry of Eu^3+^, it has been reported that the luminescence intensity due to the ^5^D_0_→^7^F_4_ transition is stronger than that of ^5^D_0_→^7^F_2_ [21]. The results shown in Figure 2 and Figure 4d may indicate that the PL intensity of Eu^2+^ or Eu^3+^ doped in BaSiO_3_ depends on the co-doped element or the environment around the Eu^2+^ or Eu^3+^, rather than the charge state of the co-doped elements such as R^+^, La^3+^, and Y^3+^.

## 4. Conclusions

In summary, Ba_0_._98+0_._02R_SiO_3_: 0.02Eu^(2+/3+)^, R^+/3+^(R = Li^+^, Na^+^, K^+^, La^3+^ and Y^3+^) phosphors were prepared by means of a solid-state reaction method and studies were carried out on the effect of co-doping with ions such as Li^+^, Na^+^, K^+^, La^3+^, and Y^3+^ on the optical properties of the Ba_0_._98_SiO_3_: 0.02Eu^(2+/3+)^ phosphors.

One notable result of this work is that the excitation efficiency due to CTB was enhanced by co-doping of R^+/3+^, which acted as a kind of sensitizer, and thus increased the emission intensity due to Eu^3+^ by 3.7 times. However, the co-doped R^+/3+^ ions did not increase the emission intensity of Eu^3+^ via direct ^7^F_0_→^5^L_6_ excitation of Eu^3+^, but rather decreased it. On the other hand, the co-doped R^+^ ions did not enhance the emission intensity of Eu^2+^ via the excitation of the CTB or through a direct excitation of Eu^2+^, but the co-doped La^3+^ did achieve this effect. The present work thus reveals for the first time that the optical properties of Eu^2+^ or Eu^3+^ doped in BaSiO_3_ depend on the co-doped element itself, rather than the charge state of the co-doped elements such as R^+^, La^3+^, and Y^3+^.

## Figures and Tables

**Figure 1 materials-15-06559-f001:**
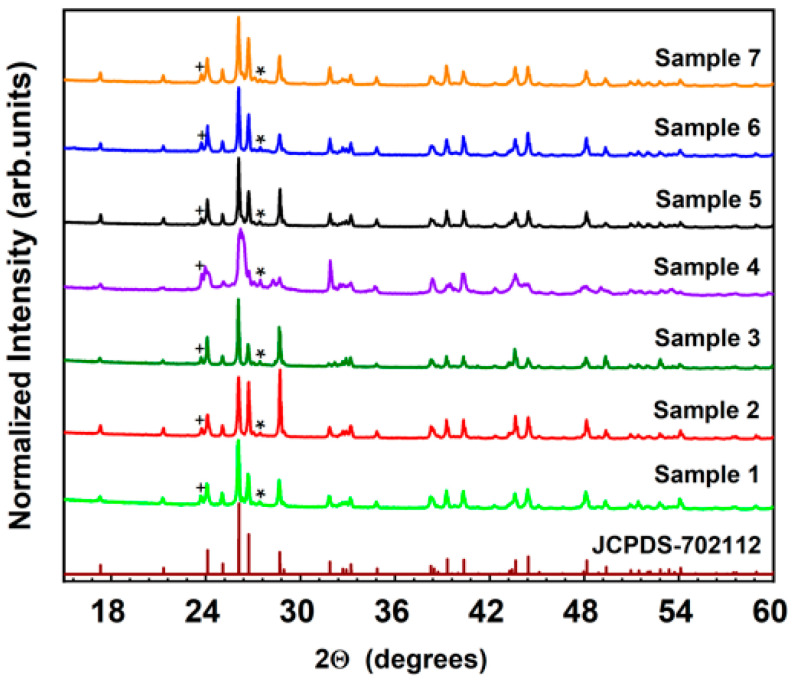
XRD patterns of Ba_0_._98_SiO_3_:0.02Eu^3+^, 0.02R (R = Li^+^, K^+^ and Na^+^).

**Figure 2 materials-15-06559-f002:**
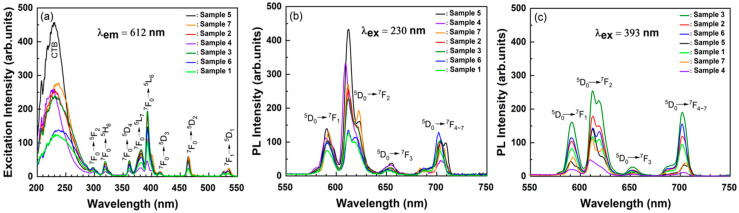
Excitation spectra (**a**) obtained by monitoring the emission at 612 nm, emission spectra at an excitation wavelength of (**b**) 230 nm and (**c**) 393 nm for Ba_0_._96_SiO_3_:0.02Eu^3+^, 0.02R^+/3+^ phosphors (samples 1–7).

**Figure 3 materials-15-06559-f003:**
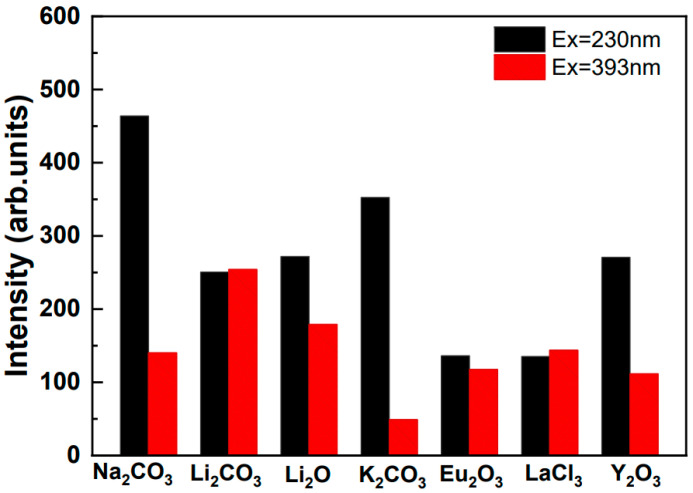
Effect of co-doping on the PL intensity of Eu^3+^ ions excited by 230 nm and 393 nm, respectively.

**Figure 4 materials-15-06559-f004:**
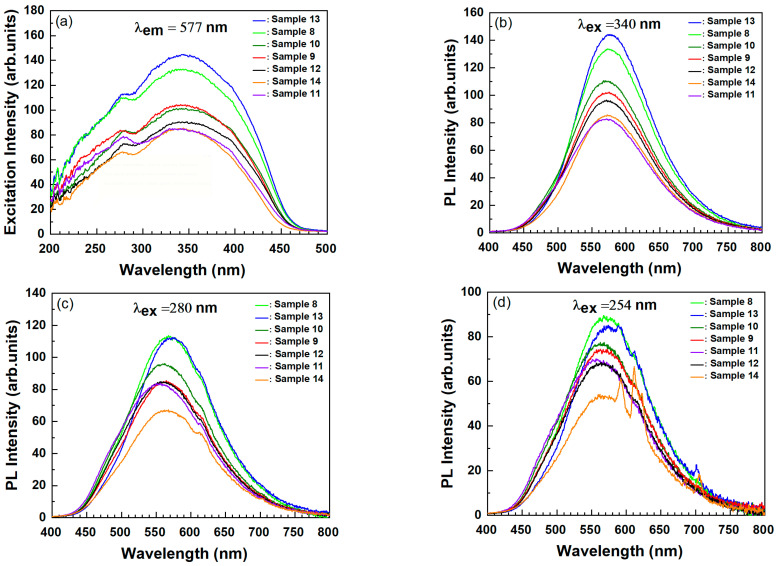
Excitation (**a**) and emission (**b**–**d**) spectra of the Ba_0_._96_SiO_3_:0.02Eu^2+^, 0.02R^+/3+^ phosphors (samples 8–14).

**Table 1 materials-15-06559-t001:** Studied samples.

Sample No	Eu^(3+/2+)^ (2%)	R	Remark
1	Eu^3+^	0	1st heat treated in air
2	Eu^3+^	0.02 Li^+^	R = Li, Li_2_O used as co-doping material, 1st heat treated in air
3	Eu^3+^	0.02 Li^+^	R = Li, Li_2_CO_3_ used as co-doping material, 1st heat treated in air
4	Eu^3+^	0.02 K^+^	R = K, K_2_CO_3_ used as co-doping material, 1st heat treated in air
5	2 Eu^3+^	0.02 Na^+^	R = Na, Na_2_CO_3_ used as co-doping material, 1st heat treated in air
6	Eu^3+^	0.02 La^3+^	R = La, LaCl_3_ used as co-doping material, 1st heat treated in air
7	Eu^3+^	0.02 Y^3+^	R = Y, Y_2_O_3_ used as co-doping material, 1st heat treated in air
8	Eu^2+^	0	same as sample 1 except 2nd heat treated in a reducing atmosphere
9	0 Eu^2+^	0.02 Li^+^	same as sample 2 except 2nd heat treated in a reducing atmosphere
10	Eu^2+^	0.02 Li^+^	same as sample 3 except 2nd heat treated in a reducing atmosphere
11	Eu^2+^	0.02 K^+^	same as sample 4 except 2nd heat treated in a reducing atmosphere
12	Eu^2+^	0.02 Na^+^	same as sample 5 except 2nd heat treated in a reducing atmosphere
13	Eu^2+^	0.02 La^3+^	same as sample 6 except 2nd heat treated in a reducing atmosphere
14	Eu^2+^	0.02 Y^3+^	same as sample 7 except 2nd heat treated in a reducing atmosphere

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
