# Peer review of "Co-Doping Effect on the Optical Properties of Eu^(2+/3+)^ Doped in BaSiO_3"

_materials, 2022, doi:10.3390/ma15196559_

Round 1

Reviewer 1 Report

This manuscript reported co-doping effect on the optical properties of Eu(2+/3+) doped in BaSiO3. In the past work, the co-doping effect of R+ on the luminescence properties of Eu3+ in BaSiO3 was evaluated in Ref.7, but in the work, the effect of R+/R3+ ion on the luminescence properties of Eu(2+/3+) doped in BaSiO3 were studied. In addition, luminescence properties were well discussed, so the manuscript should be suitable for Materials. The reviewer recommends minor revisions as follows.

1.       Please show the experimental method of sintering in a reduction atmosphere.

2.       To show appearance of samples is preferred because luminescence intensity is affected by sample size if you can.

3.    What is the possible origin of the minor peaks in XRD. 

4.       Red to Orange ratio (R value) value of Eu3+ should help the authors discuss the effect as shown in Ref.7 (Optical Materials, 2021, 114, 110981).

5.       In page 9, the author said that "in general, it is known that the emission intensity due to …… than Ω2 in the JO theory [21]. However, depending on the symmetry of Eu3+…". The sentence is unclear in the present form. Please explain in detail. At first, in some materials, Ω4 is higher than Ω2. Further, what the authors want to mention about the symmetry of Eu3+? In addition, the surrounding environment of Eu3+ has much effect on R value and Ω2, but the symmetry is no relationship with Ω4.

Author Response

We are very thankful for the reviewers’ positive comments. Suggestions to improve the manuscript have been considered and the responses are given below. The same has been incorporated in the revised version of the manuscript in appropriate places. 

This manuscript reported co-doping effect on the optical properties of Eu(2+/3+) doped in BaSiO3. In the past work, the co-doping effect of R+ on the luminescence properties of Eu3+ in BaSiO3 was evaluated in Ref.7, but in the work, the effect of R+/3+ ion on the luminescence properties of Eu(2+/3+) doped in BaSiO3 were studied. In addition, luminescence properties were well discussed, so the manuscript should be suitable for Materials. The reviewer recommends minor revisions as follows.

  1. Please show the experimental method of sintering in a reduction atmosphere.

→ The BaSiO3:Eu2+ samples were prepared by sintering again the BaSiO3:Eu3+ samples under a reduction atmosphere (95% N2 + 5% H2) at 1,200℃ for 6 hours.

  1. To show appearance of samples is preferred because luminescence intensity is affected by sample size if you can.

→ As mentioned below (page 2), it is difficult to do additional experiments.

  1.  What is the possible origin of the minor peaks in XRD. 

Authors tried to identify the miscellaneous minor peaks marked with plus (+) and asterisk (*) but could not.

  1. Red to Orange ratio (R value) value of Eu3+ should help the authors discuss the effect as shown in Ref.7 (Optical Materials, 2021, 114, 110981).

As shown in reference 7, Zhaoyun Yang et al., prepared the samples by using the conventional chemical co-precipitation method different from our solid state reaction method and the precursors were also calcined in air. They found the self-reduction effect but we could not find the self-reduction by calcining in air. Thus, we did not discuss the red to orange ratio (R value).

  1. In page 9, the author said that "in general, it is known that the emission intensity due to …… than Ω2 in the JO theory [21]. However, depending on the symmetry of Eu3+…". The sentence is unclear in the present form. Please explain in detail. At first, in some materials, Ω4 is higher than Ω2. Further, what the authors want to mention about the symmetry of Eu3+? In addition, the surrounding environment of Eu3+ has much effect on R value and Ω2, but the symmetry is no relationship with Ω4.

This is a very good point. Authors knew that the emission intensity due to the 5D07F4-7 transition was weaker than the emission intensity caused by the 5D07F2 transition because Ω4 is much smaller than Ω2 according to the Judd-Ofelt theory. However, the PL intensity of the 5D07F2 transition of Eu3+ for sample 6 co-doped with La3+ is weaker than that of the 5D07F4-7 transition. Frankly speaking, authors do not know why the PL intensity of the 5D07F2 transition of Eu3+ for sample 6 co-doped with La3+ is weaker than that of the 5D07F4-7 transition. To explain this experimental result, authors searched references and realized from ref 22 that the luminescence intensity due to the 5D07F4 transition can be stronger than that of 5D07F2 depending on the symmetry of Eu3+. Therefore, the following sentence, ‘However, depending on the symmetry of Eu3+, it has been reported that the luminescence intensity due to the 5D07F4 transition is stronger than that of 5D07F2 [22]’ was added in the0020manuscript to help readers understand.

Reviewer 2 Report

In this work, the authors have investigated the effect of a few co-doping ions on the optical properties of BaSiO3:Eu (2+/3+), and it was concluded that the optical properties are dominantly dictated by the element of co-doping than the charge state. A major revision is required.

·      The English language must be revised

·      TEM images of samples and DLS data need to be added to the manuscript

·      Information about the quantum efficiency of the samples is required

·      What is the effect of the doping ions on the PL lifetime? Add this information.

·      CIE chromaticity diagram of the samples is required

·      Figure captions need to be corrected. For instance, in figure 2 all parts need to be explained separately.

·      What was the power of the excitation lights in figure 3?

·      More recent references are required

·      Avoid using Wikipedia and online files for your reference such as 19 and 20

Author Response

We are very thankful for the reviewers’ positive comments. Suggestions to improve the manuscript have been considered and the responses are given below. The same has been incorporated in the revised version of the manuscript in appropriate places. 

In this work, the authors have investigated the effect of a few co-doping ions on the optical properties of BaSiO3:Eu (2+/3+), and it was concluded that the optical properties are dominantly dictated by the element of co-doping than the charge state. A major revision is required.

  1. The English language must be revised

→ / English corrections are made by a native speaker and all the grammatical errors are corrected in the revised manuscript now.

  1. TEM images of samples and DLS data need to be added to the manuscript.

→ Authors have recently studied the optical properties of Eu2+/3+ doped in a silicate-based matrix. While conducting research, it was found that the optical properties of Eu2+/3+ doped in BaSiO3 were different for each researcher. We realized that the optical properties of Eu3+ were mostly explained based on the charge compensator. The purpose of this study was to investigate the differences in these research results and  clarify whether the explanation based on charge compensation alone was reasonable. Therefore, TEM and DLS experiments were not performed. Co-researcher, Purevdulam Namkhai, received her Ph.D and returned to her motherland in June, so adding TEM and DLS data to the manuscript by performing additional experiments is difficult. Since the submission period for a revised manuscript is 10 days, it is very difficult to revise the manuscript to include additional experiment results within this timeframe.

  1. Information about the quantum efficiency of the samples is required.

→ As mentioned above, the goal of this research is not to improve optical properties such as emission efficiency. At present, it is difficult to measure the quantum efficiency of the samples and add it to the manuscript.

  1. What is the effect of the doping ions on the PL lifetime? Add this information.

→ In my lab, it is possible to measure the PL lifetime of Eu3+ but not possible to measure the PL lifetime of Eu2+. Thus it is difficult to add this information into the revised manuscript.

  1. CIE chromaticity diagram of the samples is required.

→ Authors think that the CIE chromaticity diagram of the samples is not useful for the goal of this research.

  1. Figure captions need to be corrected. For instance, in figure 2 all parts need to be explained separately.

 → As per your suggestion, all parts in figure 2 were explained separately below.

Fig. 2. Excitation and emission spectra of the Ba0.96SiO3:0.02Eu3+, 0.02R+/3+ phosphors (sample 1~7).

→ Fig. 2. Excitation spectra (a) by monitoring the emission at 612 nm, emission spectra at excitation wavelength of (b) 230 nm and (c) 393 nm for Ba0.96SiO3:0.02Eu3+, 0.02R+/3+ phosphors (sample 1~7).

  1. What was the power of the excitation lights in figure 3?

The photoluminescence excitation (PLE) and emission (PL) spectra were measured using a fluorescent spectrofluorometer (JASCO FP-8500, Japan) equipped with an integrating sphere (ISF-834) as explained in the materials and methods section. For reference, the xenon lamp power of FP-8500 fluorescent spectrofluorometer is 150 W.

  1. More recent references are required

→ We tried to find recent references on the Eu(2+/3+) doped BaSiO3 but it was difficult to find except those already referenced in this manuscript.

  1. Avoid using Wikipedia and online files for your reference such as 19 and 20.

→ As per your suggestion, the Wikipedia and online files for reference were deleted.

Round 2

Reviewer 2 Report

The reviewer believes the suggested corrections are required and since the authors refused or could not perform key items of 2, 3, 4, 5, and 6, the decision is to reject the paper. In the case of item 6, Figure 2 is corrected by Figure 4 remained untouched.